# Understanding variation in unplanned admissions of people aged 85 and over: a systems-based approach

Andrew Wilson,[1] Richard Baker,[1] John Bankart,[1] Jay Banerjee,[2] Ran Bhamra,[3] Simon Conroy,[4] Stoyan Kurtev,[1] Kay Phelps,[5] Emma Regen,[5] Stephen Rogers,[6] Justin Waring[7]

[1]Health Sciences, University of Leicester, Leicester, UK
[2]Emergency Medicine, University Hospitals of Leicester NHS Trust, Infirmary Square, Leicester, UK
[3]WolfsonSchool of Mechanical, Electrical & Manufacturing Engineering, Loughborough University, Loughborough, UK
[4]Health Sciences, University of Leicester, Leicester, UK
[5]Health Sciences, University of Leicester, Leicester, UK
[6]Health Sciences, University of Leicester, Leicester, UK
[7]Centre for Health Innovation, Leadership & Learning / Nottingham University Business School, University of Nottingham, Nottingham, UK

**Correspondence to**
Professor Andrew Wilson;
aw7@le.ac.uk

## ABSTRACT

**Aim** To examine system characteristics associated with variations in unplanned admission rates in those aged 85+.

**Design** Mixed methods.

**Setting** Primary care trusts in England were ranked according to changes in admission rates for people aged 85+ between 2007 and 2009, and study sites selected from each end of the distribution: three 'improving' sites where rates had declined by more than 4% and three 'deteriorating' sites where rates had increased by more than 20%. Each site comprised an acute hospital trust, its linked primary care trust/clinical commissioning group, the provider of community health services and adult social care.

**Participants** A total of 142 representatives from these organisations were interviewed to understand how policies had been developed and implemented. McKinsey's 7S framework was used as a structure for investigation and analysis.

**Results** In general, improving sites provided more evidence of comprehensive system focused strategies backed by strong leadership, enabling the development and implementation of policies and procedures to avoid unnecessary admissions of older people. In these sites, primary and intermediate care services appeared more comprehensive and better integrated with other parts of the system, and policies in emergency departments were more focused on providing alternatives to admission.

**Conclusions** Health and social care communities which have attenuated admissions of people aged 85+ prioritised developing a shared vision and strategy, with sustained implementation of a suite of interventions.

## INTRODUCTION

Internationally, unplanned hospital admissions have increased steadily over recent decades.[1–4] In England, between 2001/2002 and 2012/2013, unplanned admissions of people aged 65+ rose by 46% and the age-standardised rate by 25%. Rates of increase rose steadily with age: from 9.9% for those aged 65–69% to 50.2% for those aged 90+.[5]

Research consistently finds unexplained variations in the rise of unplanned

### Strengths and limitations of this study

► In England, unplanned hospital admissions for people aged 85 and over are rising but there is substantial geographical variation.
► A 'whole system approach' can be used to understand this variation between health economies.
► Through qualitative interviews with a comprehensive range of informants, we examined three sites where rates of admission in this age group rose most sharply and three in which the rise was reversed or attenuated.
► The study relied on institutional memory at a time of transition.

admissions, suggesting lessons may be learnt from different experiences.[6–9] Alternatives to admission are particularly important in those aged 85+, who often present with multiple comorbidities, polypharmacy, cognitive impairment and disability. Once admitted, they have longer stays, are more prone to hospital acquired complications and may experience more difficulty returning to their usual place of residence.[10 11] Furthermore, minimising time spent in hospital is a health outcome that matters to older people.[12] There is increasing evidence that alternatives to acute admission, such as Hospital at Home, produce similar if not better outcomes,[13] although in England, provision of these services remains about half what is needed.[14]

Several initiatives have been introduced to address the increase in acute admissions.[6] There is good evidence that higher continuity in primary care is associated with fewer admissions,[15] and that senior review in emergency departments (ED) can be effective in reducing admissions.[16] There is also some evidence of benefit from integrating primary and secondary care and health and social care.[17] However as the Kings Fund report concluded, 'a combination of interventions

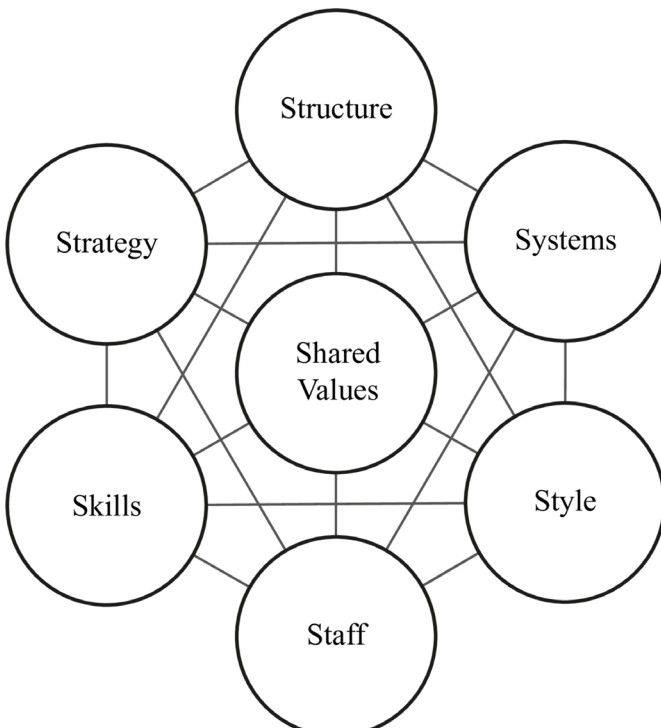

**Figure 1** The McKinsey 7S framework and how it was applied. Strategy: the plan of activity for the whole system, and alignment of the system to its goals. Structure: how different components of the system related to each other. Systems: individual services contributing to the whole system. Shared values: the norms and standards that guide the behaviour of the human elements within the system. Style: the style of management used by the system leadership. Staff: training, motivation and rewards of the staff. Skills: specific skills existing and required by staff in order to best execute their duties.

intended to reduce admissions may be expected to have a 'cumulative' effect'.[6] To understand variations in admissions it is therefore useful to examine the configuration of the whole system.

As most of the variation in admission rates is due to relatively immutable demographic factors such as deprivation,[6] examining differing trends in rates over time provides an opportunity to explore the impact of changes in service configurations.

In this study, we aimed to identify system characteristics associated with higher and lower increases in unplanned admission rates in those aged 85+. This article focuses on the qualitative component of a mixed method study reported fully elsewhere.[18]

## METHODS
### Methodology
We used a qualitative, multiple, explanatory case study approach as our principal method.[19] Following consideration of tools to enable the analysis of complex organisations, the McKinsey 7S framework was selected as shown in figure 1.[20] Its particular contribution was to help understand the inherent complexity of the system as a whole, and emphasise that for change to be effective, changes in any one component should be accompanied by complementary changes in others.

### Site selection
Six study sites were selected based on admission data for patients aged 85+ from English primary care trusts (PCTs). These organisations were responsible for commissioning primary, community and secondary health services, and were succeeded by clinical commissioning groups (CCGs) in 2013. We used PCTs as the basis of site selection as these had a population base to derive admission rates.

Admission rates for people aged 85+ were calculated from HES data (2007/2008–2009/2010).[20] Data were not available for some PCTs due to mergers. For the 143/154 (93%) PCTs for which we had data, a regression coefficient was calculated for the change in admission rates over the 3-year period, adjusting for population size and age, which was used to rank PCTs.

After applying our inclusion and exclusion criteria (see below), we selected the three sites where rates of admission had increased most rapidly, and the three where they had declined most rapidly. Each site comprised an acute hospital trust, its linked PCT, the provider of community health services and adult social care. An inclusion criterion was that more than 80% of acute admissions for people aged 85+ from the PCT were admitted to one acute Trust, so that there was at least a potential partnership between these organisations. Sites were excluded if they were known to be experiencing significant reconfiguration as reflected in publicly available information. Table 1 shows the ranking and admission rates for the selected sites: improving sites had an annual decrease in admission rates for those aged 85+ of 1%–2%, and deteriorating sites an annual increase of 6%.

### Quantitative data collection
After the sites had been selected, we used updated HES data to examine 85+ admission rates over 5 years (2007/2008 to 2011/2012) rather than just the 3 years used for selection. We used the NHS portal (now NHS digital[21]) to examine admissions for acute and chronic ambulatory care sensitive conditions (ACSCs)[22] from 2007/2008 to 2009/2010. These are age-adjusted rates per 100 000; data are not available for specific age groups. In the selected sites, invitations to participate were sent to the chief executives of the PCT and acute trust. In all cases, there was initial agreement from both parties. We then invited participation from the organisation responsible for community health services and social services.

### Qualitative data collection
In each selected site, a key individual was identified in each organisation who advised on potential key informants. Further participants were identified by snowball sampling.[23] In 2013, two rounds of data collection were conducted. In the first round, an understanding of the

**Table 1** Selection of improving (I) and deteriorating (D) primary care trusts (PCT)

| PCT Rank for slope (n=143) | 85+ admission rate (number of admissions/100 population aged 85+) | | | Slope (per annum change) | % change in rate | % admissions to linked hospital trust | % aged 85+ | Reference in paper |
|---|---|---|---|---|---|---|---|---|
| | 2007/08 | 2008/09 | 2009/10 | | | | | |
| 4 | 0.55 | 0.51 | 0.51 | −0.02 | −7.3 | 89 | 2.6 | I1 |
| 5 | 0.61 | 0.6 | 0.57 | −0.02 | −6.6 | 87 | 2.6 | I3 |
| 9 | 0.41 | 0.41 | 0.39 | −0.01 | −4.9 | 83 | 2.2 | I2 |
| 132 | 0.48 | 0.54 | 0.59 | 0.06 | 22.9 | 92 | 2.2 | D1 |
| 133 | 0.41 | 0.45 | 0.52 | 0.06 | 26.8 | 87 | 1.7 | D3 |
| 135 | 0.49 | 0.59 | 0.61 | 0.06 | 25.5 | 83 | 1.8 | D2 |

system's history and drivers was sought in interviews with key high-level informants, including commissioners and managers of health and social care with responsibility for those aged 85+, and clinicians and care providers with leadership roles in primary care, ED, social care, intermediate and secondary care. In the second round, we examined specific components of the system, using in-depth interviews and focus groups with those involved in delivering care, to explore issues involved in translating policy directives to changes in provision of care. These included clinicians in ED and acute medical units (AMUs), managers of intermediate and integrated care provision and clinicians in primary care. In each site, a focus group was convened including representatives of carers and service users to capture their perspectives.

Topic guides were based on the literature and agreed by the research team. The structure of the topic guide was as follows: views on unplanned admissions, views on system characteristics, specific questions on system characteristics, changes and recommendations, patient public involvement (PPI), outcomes. A detailed topic guide is presented in online supplementary appendix 1.

For each site, data on population characteristics and admission rates were prepared to prompt discussion. Interviews were conducted at the workplace by research fellows (ER and KP) with substantial experience of policy-focused qualitative work, who also made extensive fields notes for each site. Interviews and focus groups lasted 30–60 min and were recorded and transcribed verbatim. Data were collected between January 2012 and December 2013.

### Analysis

Qualitative data analysis was undertaken in a stepwise, interpretative approach. First, all data from each case site were assigned to individual members of the project team for initial inductive, open coding. The second and main stage of data analysis involved two independent researchers developing detailed case reports. Following a framework approach,[24] all data items were systematically scrutinised with data coded according to the 7S categories. These codes and descriptions provided the

basis of regular discussion among the multidisciplinary research team, first, for reviewing the consistency of the analysis process (intercoder), that is, so that similar codes related to similar phenomena; and then for debating and agreeing the thematic interpretation of data. As part of this process, the similarities and differences between codes were analysed, especially when relating data the 7S model, to ensure that data/codes were categorised in ways that were sufficiently distinct, or where they shared common or complementary features they were aggregated into higher order codes, which was especially important when relating the data to the McKinsey model. NVivo software[25] was used to provide an audit trail. Guidance for coding was agreed by the team, including how items would be categorised according to the 7S framework. In line with the principle of constant comparison, each category was systematically checked for its internal consistency and inter-relationships.[26] Through the processes of analysis and interpretation, the research team was especially concerned to test the reliability and confidence of interpretation through looking for counterfactuals in the data that could represent contingencies to the emerging interpretations. Illustrative and exemplar extracts of data are provided in the subsequent results section, and further empirical data can be found in the main study report.

### Patient and public involvement

The study was presented to the Leicester older people's research PPI forum at the planning stage, and a representative from this forum was a member of the steering group.

NHS ethical approval was not required as patients were not being interviewed. Ethics approval was obtained. Informed consent was obtained from all participants.

### RESULTS
### Quantitative findings

Table 2 presents data on rates of 85+ admissions between 2007/2008 and 2011/2012. In improving sites, these increased by 3.3% (range −2.4% to 10.6%) and in deteriorating sites by 22.7% (range 16.3% to 29.2%). During

**Table 2** Changes in admission rates for patients aged 85+, and rates of admission for acute and chronic ambulatory care sensitive conditions (ACSC) in improving and deteriorating sites

| | I1 | I2 | I3 | D1 | D2 | D3 | Improving sites average* | Deteriorating sites average* |
|---|---|---|---|---|---|---|---|---|
| **85+ admissions/100 aged 85+ per annum** | | | | | | | | |
| 2007/2008 | 0.47 | 0.41 | 0.57 | 0.48 | 0.49 | 0.40 | 0.48 | 0.46 |
| 2008/2009 | 0.51 | 0.42 | 0.6 | 0.55 | 0.59 | 0.45 | 0.51 | 0.53 |
| 2009/2010 | 0.51 | 0.40 | 0.58 | 0.60 | 0.61 | 0.53 | 0.50 | 0.58 |
| 2010/2011 | 0.54 | 0.39 | 0.58 | 0.60 | 0.56 | 0.48 | 0.50 | 0.55 |
| 2011/2012 | 0.52 | 0.40 | 0.58 | 0.62 | 0.57 | 0.49 | 0.50 | 0.56 |
| % change between 2007/2008 and 2011/2012 | 10.64 | −2.44 | 1.75 | 29.17 | 16.33 | 22.50 | 3.32 | 22.67 |
| Linear regression slope (per annum change in rate) | 0.013 | −0.005 | 0 | 0.033 | 0.013 | 0.020 | 0.003 | 0.022 |
| **Acute ACSC all ages. Indirectly age and sex standardised rate per 100000** | | | | | | | | |
| 2007/2008 | 424 | 579 | 462 | 367 | 397 | 619 | 488 | 461 |
| 2008/2009 | 515 | 591 | 347 | 388 | 461 | 699 | 484 | 516 |
| 2009/2010 | 350 | 627 | 387 | 449 | 416 | 785 | 455 | 550 |
| % change between 2007/2008 and 2009/2010 | −17.45 | 8.29 | −16.23 | 22.34 | 4.79 | 26.82 | −8.46 | 17.98 |
| Linear regression slope (per annum change in rate) | −37 | 24 | −37.5 | 41 | 9.5 | 83 | −16.83 | 44.50 |
| **Chronic ACSC all ages. Indirectly age and sex standardised rate per 100000** | | | | | | | | |
| 07/08 | 220 | 240 | 148 | 188 | 177 | 247 | 203 | 204 |
| 08/09 | 276 | 247 | 104 | 212 | 177 | 265 | 209 | 218 |
| 09/10 | 210 | 241 | 137 | 218 | 152 | 249 | 196 | 206 |
| % change between 2007/2008 and 2009/2010 | −4.55 | 0.42 | −7.43 | 15.96 | −14.12 | 0.81 | −3.85 | 0.88 |
| Linear regression slope (per annum change in rate) | −5 | 0.5 | −5.5 | 15 | −12.5 | 1 | −3.33 | 1.17 |

*Averages derived from rows not columns.

the 5-year period, the mean linear regression slope per annum was 0.003 (0.3%) for improving sites and 0.022 (2.2%) for deteriorating sites. In general, rates of admission remained fairly stable in the additional 2 years we examined after site selection. Between using HES data for selection and this analysis, some corrections had been made to 2007/8 data.

Changes in rates of admission for ACSCs (acute conditions such as otitis media that could normally be managed without admission and chronic conditions, such as diabetes, in which disease management could prevent exacerbations needing admission[8]) are also shown in table 2. In the 3 years examined, rates for acute conditions fell by 8.5% in improving sites (range −17.5% to 8.3%) and increased by 18.0% in deteriorating sites (range 4.8% to 26.8%). Mean rates of change differed less for chronic conditions, but with bigger variations between sites. They reduced by 3.9% in improving sites (range −7.4% to 0.4%) and increased by 1.0% in deteriorating sites (range −14.1% to 16.0%).

### Qualitative findings

In total, 142 informants contributed to either individual interviews or focus groups, as shown in table 3, which also provides a brief description of the sites. The number of contributions from each site varied, in part because some agencies declined to participate. In site I2, community services and social services were provided by a single organisation. The number of informants was greater in improving than deteriorating sites (91 vs 51)

Findings are presented by 7S themes, as summarised in table 4. Quotes are identified by site and employing organisation of respondent (PCT/CCG: A; community services: B; acute trusts: C) and respondent number. All quotes presented are from one-to-one interviews.

### Strategy

This was defined as the plan of activity for the whole system, and alignment of the system to its goals. Improving sites exhibited more of a shared and comprehensive system-wide strategy for managing unplanned care, including specific policies and procedures for older people. These strategies and policies were shared across the wider health and social care system suggesting an underlying basis of collaboration and coordination, and a reduced risk of system dominance by one provider.

> We had an audacious programme goal, which was all about reducing emergency admissions, and we had quite a lot of buy-in … to a whole-system approach. I1 A-01 (commissioning manager, CCG)

> For the whole time I've been here, it (urgent care) has been a top priority. We want urgent care pathways (including ambulance, ED and admissions) to be as high quality as possible. I3-A-02 (commissioning manager, CCG)

Deteriorating sites revealed less evidence of a system strategy. Although individual system components had

developed strategies for aspects of unplanned care, such as reducing length of stay, there was less appreciation of how the components of the wider health system should work together. Strategies tended to be dominated by acute care provision to the detriment of policies to expand primary and community care.

> The system plans have bullets like 'we'll support care close to home but we'll support financial sustainability of the acute hospitals.' Unless you have some sort of integration, those two things are mutually exclusive in the long term. D1-A-03 (service redesign manager, CCG)

An important difference between improving and deteriorating sites was their approach to improvement projects. In improving sites, these were generally well resourced, often through funding arrangements linked to national initiatives. Moreover, they were usually given time to develop and embed into practice, rather than being subject to changing fashions or emerging policies. In contrast, in deteriorating sites projects tended to be more reactive and short-lived with little follow through.

> We've piloted lots of good things, but it's been the usual story of just doing pilots and not doing them at sufficient scale, we've dabbled in things and haven't really followed them sufficiently through. D3-A-03 (commissioning director, CCG)

### Structure

This was defined as how different components of the system related to each other. In improving sites, there was closer integration of primary, acute and community services. This was facilitated by fewer organisations providing services and clearer geographical boundaries.

> X was one of the few dedicated Community Trusts in the country…there's not several different providers. Patients can't get moved around the system because there's only basically GP practices, intermediate tier and one acute trust. I1-B-01 (rehabilitation manager, community trust)

> The care trust… was integrated – the health organisation held the social care budget on behalf of the council. …that alone meant that the approach to commissioning was truly integrated at the budget and organisational level. So that I think is fundamental. I2-A-01 (service lead, unplanned care, CCG)

> We hardly have any cross-boundary issues, it's just one social services department, one acute trust, one community trust, one mental health trust and one health commissioner – that's it. I3-A-02 (commissioning manager, CCG)

In deteriorating sites, there was less evidence of integration between acute, primary and community services. Governance and funding arrangements were more complex, with different ways of working. There was

**Table 3** Description of sites and informants by organisational category

| Sites | Description | | | | | Participants | | | | | |
|---|---|---|---|---|---|---|---|---|---|---|---|
| | ONS classification | Ranking of population size (151 PCTs, 1=largest) | N (%) aged 85+ | Deprivation rank (151 PCTs, 1=most deprived) | Acute provision | Acute Trust | PCT/CCG | Community services | Social services | PPI | Total |
| I1 Major city | Regional centre | 85 | 6527 (2.6) | 56 | One university hospital, one district general hospital | declined | 2 | 13 (seven individual, two focus groups (n=2 and 4)) | Declined | | 15 |
| I2 Largely rural area comprising three small to medium-sized towns. | Manufacturing town | 139 | 3546 (2.6) | 40 | Three district general hospitals | 6 | 6 | 16 (seven individual, two focus groups (n=9)) | Declined | Focus group (n=5) | 33 |
| I3 Semirural and urban conurbation in close proximity to metropolitan area. | Industrial hinterlands | 42 | 7970 (2.2) | 50 | Two district general hospitals | 5 | 3 | 24 (four individual, three focus groups (n=6,6,8)) | 2 | Focus group (n=9) | 43 |
| D1 Major city | Centre with industry | 56 | 6667 (2.2) | 43 | One large acute hospital | 7 | 3 | 3 | 2 | Focus group (n=5) | 20 |
| D2 Three small to medium-sized towns | Centre with industry | 120 | 3703 (1.7) | 22 | Four district general hospitals | 10 | 3 | 2 | 4 | Focus group (n=5) | 24 |
| D3 Mixed urban and rural area | New and growing town | 118 | 3463 (1.8) | 119 | One large acute hospital | 2 | 3 | Declined | 1 | 1 | 7 |
| Total | | | | | | 30 | 20 | 58 | 9 | 25 | 142 |

**Table 4** Features of sites by McKinsey 7S categories

| 7S | Strategy | Structure | Systems | Shared values | Skills | Style | Staff |
|---|---|---|---|---|---|---|---|
| I1 | High levels of investment in community provision. Early engagement of all stakeholders in strategy development | Strong linkages between hospital, GP and community care. Small number of large providers | Innovative cross-sectoral technology systems. Unified provision of intermediate care | Professionals willing to work together and bend hierarchies to reach shared goals. Pride in providing services to keep older people out of hospital | Staff have perseverance and skills to see through projects and get people on board | Regular contacts between hospital and community providers | Effective multidisciplinary teams in intermediate care. Recognised need to involve pharmacists to reduce excessive medication |
| I2 | Care trust developed strategy across social and community health services | Integrated budget for commissioning health and social care | Focus on practice-based commissioning providing incentives for GPs to reduce admissions. Single point of access 24/7 for intermediate care | Strong and stable organisational cohesion | High levels of interpersonal skills enable effective working relationships | Close links between GPs and other service providers | Longstanding and close working relationships |
| I3 | Urgent care a top strategic priority. Strategy recognises the importance of community care | Lack of boundary issues helps maintain a clear structure, with small number of providers | Out of hours run by through community trust. Rapid access to clinics and telephone consultations with geriatricians | Strong organisational cohesion | High levels of skills in community teams | Some 'blame culture' when services are pressurised | |
| D1 | Frequent changes in leadership roles and regional strategies. Acute trust dominant in determining strategy | Complex structures for health community care | Perception that GPs were demotivated. Lack of well-functioning multidisciplinary teams | Historically poor relationship between trusts and community providers | Perception that care home staff are underskilled | Culture of admission from ED as default option | Recognised shortage of geriatricians |
| D2 | Lack of clear strategy on unplanned admissions; more focus on reducing length of stay than admission avoidance | Poor integration between primary and secondary care, and between ambulance services and acute trust | Frequent restructuring of intermediate care | Lack of shared culture between organisations | Recognised need for skilled geriatricians in acute medical unit | Recent focus on clinical leadership | Lack of senior medical staff in ED. Recent investment in geriatricians and community matrons |
| D3 | More focus on elective care than urgent care. Successful pilots not followed through | Frequent changes in structure of system, including hospital sites and structure of intermediate care | Low investment in primary care | Conflict between medical, rehabilitation and managerial values | Perception that insufficient staff have skills needed to assess frail elderly | Recent focus on clinical leadership | Inadequate provision of community matrons |

often emphasis on key care stages, such as admission and discharge, but not on the wider constellation of agencies, handovers and transitions that patients face across the system.

> The services are so complex that we as health professionals find it difficult and I think … for somebody who is over 85 or for anybody I think - it's really difficult for them to understand where to go for help and I think so by default people know that this (A and E) is open 24 hours a day, you can literally just walk in and you'll be helped. D1-C-01 (service coordinator for older people, acute trust)

> I don't think it [unplanned admissions], is managed very well really in this area which is one of the reasons why you see relatively high admission rates. Part of the reason for that is because there are separate organisations each with their own agendas and each with their own pressures. D2-C-01 (clinical director, unscheduled care, acute trust)

### Systems

This was defined as individual services contributing to the whole system. The most consistent differences between improving and deteriorating sites were in GP services, intermediate care and ED provision.

#### General practice

In improving sites, general practice appeared better supported financially, more innovative, for example, with IT systems, and better integrated with other providers. There was closer alignment of out-of-hours GP services with either community or acute NHS providers, which facilitated closer integration of primary, acute and community services, especially for information sharing and continuity of care.

> Round here I think GPs are more proactive in managing their patients. If you're a GMS [General Medical Services] practice there's no incentive for you keeping your admission. We had our PBC [practice-based commissioning] budgets and if we were within our budget there'd be some financial reward for that. I2-A-01 (service lead, unplanned care, CCG)

> (IT system) facilitates three-way conversations; the GP will be on the line, the nurse would be on the line and the acute physician from AMU [the acute medical unit] would be on the line and having a conference about whether or not it's appropriate that person ought to go to hospital. I1-A-01 (commissioning manager, CCG)

In deteriorating sites, there was a sense of underinvestment and insufficient planning for primary care. GP practices were seen as providing a more limited set of services with problems of access due in part to single-handed practices and half-day closures, thereby increasing demand on EDs. Changes in out of hours GP provision was also thought to impact on ED attendances.

> There's been underinvestment in primary care, there hasn't been a clear primary care strategy… there's been underinvestment in primary care local enhanced services compared to other places. D3-A-03 (commissioning director, CCG)

> We've got a high proportion of (older) GPs, it's I think over 50% who have retired, taken their pension and come back. So there is no motivation for them to change at all. D1-A-02 (head of development, CCG)

#### Intermediate care

Intermediate care provision at scale was widely seen to be important. This was more developed at improving sites but even here, short-term funding meant services were not stable. Although type of provision varied, for example, the balance of home and institutionally based provision, key elements included a single point of access, a unified system often from a single provider, and multidisciplinary working to reduce duplication.

> We have a single point of access manned by advice officers, behind which there are three levels of triage– at every level of triage it's an integrated triage between health and social care. I2-A-03 (strategic adviser, adult social care)

> The multidisciplinary teams in intermediate care have made a difference– you know, sometimes, one person's seeing a physio, an OT and a nurse, and they just need to see one member of the team. I1-B-02 (head of reablement, community trust)

#### Emergency departments

There was agreement that skill mix in ED could influence the number of older patients admitted. Initiatives in improving sites included provision of geriatricians in ED, more senior staffing and involvement of GPs. Linked to this were initiatives to provide GP with accessible alternatives to acute admission, such as rapid access clinics or telephone consultations with a geriatrician.

> GPs in A & E – well GPs have been in the hospital ever since I've been here, so they've either been co-located, so very close to A & E or in A & E. I2-A-02 (commissioning manager, CCG)

> We introduced our (older people's) clinic. If they (GPs) have got an elderly patient with them in the surgery, they're just not sure what to do, they just pick up this phone line, and there's a geriatrician who will advise them. I3-C-04 (director, acute and critical care, acute trust)

### Shared values

Shared values within and between components of the system appeared important. Improving sites were characterised by stable staffing and leadership that supported continuity of purpose, fostered trust and collaborative working and maintained commitment to improvement. In contrast, deteriorating sites had higher staff turnover

and appeared more distracted by short term changes in policy.

> It (stability) helps in terms of building those relationships and building trust and allowing us to perhaps take more risks. I2-A-02 (commissioning manager, CCG)

> I think the things that work really well are the relationships at an operational, and to a degree going into strategic work, and I think those individual relationships, and people knowing each other and having a level of trust. I3-C-02 (commissioning manager, CCG)

> No-one's in place long enough … the natural political cycle is shorter than the natural planning cycle for the health system. D3-A-01 (GP, CCG board member)

Several other values emerged from the interviews. These included a general belief that admission can be counterproductive for older people and the need to challenge a mindset that admission is the default option.

> Acute hospitals are fundamentally not the right place for over eighty fives… you're far better in your home environment or in a supported environment that's not hospital…D3-C-01 (chief executive, acute trust)

> Well, there is a culture of admission in this hospital –someone comes to the front door, they see a junior doctor, admit them. I mean, you know, if I go on a post-take ward round and see a patient that's been admitted to one of these wards on the same day and say they can be discharged, the nurse looks at me as though I'm some sort of idiot: 'What are you talking about discharge? I haven't finished admitting them yet.' D1-C-02 (clinical director, acute trust)

Several sites described a clash of values between commissioners, managers and clinicians, particularly when managing older people, and the importance of clinical leadership.

> No, there's nobody clinical, there's nobody caring, there's nobody who actually does the business of looking after people, particularly the frail elderly, who are messy and don't fit into a clear protocol. I2-A-05 (GP, CCG board member)

> I think what we probably need now is clinically led provider organisations cause you've still got managerial led provider organisations. The doctors do talk to each other but then they don't talk to the managers and the managers talk about a different thing so it's not necessarily connected. D1-A-02 (head of development, CCG)

Although the remaining 7S categories (Skills, Style, Staff) are shown separately in table 4, we felt it was clearer to present findings in these categories in the context of the four categories presented above. For example, there was overlap between style and values, and issues of skills

and staff were often raised in the context of specific system, for example, ED.

## DISCUSSION
### Summary of findings
We found some important differences between sites in which admission rates for people aged 85+ had increased most rapidly and sites in which these rates had stabilised or declined. In improving sites, there was more evidence of strong strategic leadership, enabling the development of a comprehensive systemwide strategy, including specific policies and procedures for older people, which were shared across a more integrated health and social care setting. This encouraged longer-term, consistent development of strategies, often in the face of changing national imperatives. This stability also allowed trust and shared commitment to be established and the emergence of common values across the system.

In improving sites, primary care appeared stronger, both in terms of service provision and strategic engagement. This could be one reason why admissions for ambulatory care sensitive conditions fell on average in improving sites and increased in deteriorating sites. There is also evidence presented in our funder's report[18] that GP access was better in improving sites. Intermediate care was also more developed in improving sites. These services appeared to work best when provided at scale and fully integrated with each other, offering round-the-clock availability with a single point of access, shared information systems and specialist nursing and geriatric support. Improving sites also seemed more equipped to reduce admissions of older people from ED, through a variety of initiatives, including more senior staffing, involvement of GPs and provision of specialist nurses or geriatricians. In summary, improving sites generally made fuller use of a suite of strategies to reduce unplanned admissions, and importantly had a more systemwide outlook and strategic approach.

### Comparison with previous studies
Findings from several recent qualitative and mixed methods studies are consistent with those reported here. A study examining six emergency and urgent care sites suggested that improving GP out of hours access, senior review in ED and multidisciplinary teams could reduce admissions,[27] and others have emphasised how the culture and staffing in ED can influence admission rates.[28 29] We are not aware of previous qualitative work examining the whole system, but our findings support recommendations on integrated care made by the Kings Fund, including 'sharing sovereignty', developing a persuasive vision and establishing leadership.[30] More recently, a CQC report concluded: 'To truly coordinate care, local system leaders must ensure there is a golden thread linking vision to delivery, so that everyone involved can not only share the vision but see themselves as part of the team that delivers it'.[31]

## Strengths and limitations

Our study design offered strong internal validity through in-depth analysis within case and structured comparison between cases.[32] The design and conduct of this study reflects a number of features to enhance trustworthiness.[33] The approach allowed a common method of data collection and analysis subject to open scrutiny by the wider research team and advisors, enhancing dependability. In terms of confirmability, the researchers were supported in being reflexive about their own role in data collection, including review meetings with the wider research team.

That researchers and participants knew how each site had performed helped to inform the interviews; by presenting data on admissions, we were able to engage in a more detailed discussion of issues and strategies. It is unlikely that informants would be unaware of their own performance. However, this approach may have meant that both interviewers and participants may have focused on what was perceived to be working well in improving sites, and on more negative issues in deteriorating sites. In retrospect, blinding some rounds of data analysis to categorisation of sites may have reduced the risk of bias.

The study had several other limitations. Our selection criteria were based on changes in historic data; the additional data on admissions for 2010/2011 and 2011/2012 showed that over the 5-year period, differences between improving and deteriorating sites persisted but in most sites stabilised somewhat over the last 2 years, and the trajectory of improvement or deterioration slowed. As with any study of outliers, there is a possibility that some changes represented regression to the mean. Furthermore, there were differences in performance within improving and deteriorating groups: in sites I1 and I3, the 5-year trend showed a small increase in admission rates, although less than in deteriorating sites. Some informants found it challenging to reflect on past events rather than the current situation. The strongest interview data came from informants who had been in post for the period of interest and so able to provide an institutional memory. This problem was compounded by the fact the study was conducted during a period of organisational upheaval in the NHS, leading to many informants being relatively new in post. Finally, the snowball sampling technique may have led to under-representation of some groups, such as ambulance services.

The pace of change within the NHS has quickened since the study was conducted, with additional challenges imposed by austerity. However, we feel our key messages remain relevant as they emphasise the need for strategies and stability that can weather the impact of ongoing changes. We also acknowledge that sites with a more complex mix of acute trusts (which we deliberately excluded) may experience different or additional problems and solutions.

The McKinsey 7S framework was effective in enabling the systematic investigation of system components and their interaction, but was less useful in mapping more abstract issues such as style and values. The framework appeared better suited to examining individual organisations, rather than large complex systems of interdependent heterogeneous system actors, which may have been further understood and analysed using, for example, Beer's Viable Systems Model.[22]

## Implications for policy makers and service providers

This study supports taking a whole system approach in designing services for older people. This is best achieved by developing a sustained and shared vision focusing on outcomes that matter to older people,[12] establishing strong leadership without dominance by one organisation, maximising integration and minimising complexity within the system. These findings have potential to inform the practical implementation in England of Strategic Transformation Partnerships and Accountable Care Organisations, vehicles designed to promote system-based collaboration.[34]

## CONCLUSIONS

Health and social care communities which attenuated unplanned admissions of people aged 85+ prioritised developing a shared vision and strategy, encompassing multiple organisations and backed by strong leadership and shared values. This allowed sustained implementation of a suite of interventions, including better-developed primary and intermediate care services working closely with the hospital and emergency departments.

**Acknowledgements** We are grateful for the support of members of the steering group throughout the project, and for advice from the Leicestershire, Northamptonshire and Rutland Older People's Research Patient and Public Involvement Forum. Access to sites was facilitated by the NIHR Clinical Research network. Most of all we would like to acknowledge participants at all study sites for their time and the candid insights they provided.

**Contributors** All authors contributed to study design and analysis, and have approved the final manuscript. AW led the study, RB and SR contributed expertise in primary care and quality improvement, JBank and SK led on selection of sites. JBane and SC constructed expertise on ED policies and geriatric medicine respectively. RB contributed expertise on system theory and JW led the analysis. KP and ER led on fields work and conducted most of the interviews.

**Funding** The project was funded by the NIHR Health Service and Delivery and Research Programme (project number 10/1010/05). The views and opinions expressed in this article are those of the authors and do not necessarily reflect those of the Health Service and Delivery and Research Programme, NIHR, NHS or the Department of Health.

**Competing interests** None decalred.

**Patient consent for publication** Not required.

**Ethics approval** University of Leicester and R & D approvals fromparticipating organisations

**Provenance and peer review** Not commissioned; externally peer reviewed.

**Data sharing statement** Full transcripts are available from the corresponding author.

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
