## [Reviewer comments · BMJ Open]

ARTICLE DETAILS

TITLE (PROVISIONAL)	Understanding variation in unplanned admissions of people aged 85 and over: a systems-based approach
AUTHORS	Wilson, Andrew; Baker, Richard; Bankart, John; Banerjee, J; Bhamra, Ran; Conroy, Simon; Kurtev, Stoyan; Phelps, Kay; Regen, Emma; Rogers, Stephen; Waring, Justin

VERSION 1 - REVIEW

REVIEWER	Jeffrey Fuller Flinders University Australia
REVIEW RETURNED	23-Sep-2018

GENERAL COMMENTS	DESCRIPTION OF THE STUDY Thank you for asking me to review this paper on the important and interesting topic of understanding variation of acute admissions of people aged 85 and over. The study seeks to compare six Primary Care Trust (PCT) sites in England, three that have shown reduced admissions (called improved sites) over a five-year period and three that have shown increased admissions (called deteriorating sites) over the same period. The intent was to see if there are health system factors that differ between the improving and deteriorating sites. Such differences could then be used to infer what health system factors lead to better inter-organisational care that keeps older people out of hospital. A clear strength of the study is the method used to identify the improving and deteriorating sites over the study years, whereby three sites were chosen at each end of the improvement-deterioration spectrum. STUDY DESIGN The stated method is case study, but I am unclear what are defined as the cases. Are these each of the six sites (that is six case studies) or are these the two groups of three sites (that is two case studies). I raise this as the results are presented thematically with the differentiation made cross-sectionally according to the category "improving sites" or "deteriorating sites". This would suggest two case studies. However, in describing the analysis, the authors refer to the middle analytic step of writing up six case reports, which suggest six case studies. In my view a method of using six case studies would be a more rigorous study design as this would lend itself to the use of greater contextual description of each case, including the differences in populations being served. Six site case studies would also lend itself to the search for
---

variation and disconfirming findings across sites within the improvement and deteriorating categories. At present there is very little contextual description provided other than what is provided in the selective quotes and in the overall statements made by the authors about the differences between the improving and deteriorating site categories collectively. The reader is left to accept these author descriptions at face value as accurate differentiations. The strength of case study as a research design is the detailed use of context to help explain variation, and so more detailed case context would strengthen the veracity of the claims being made in the paper.

The authors may argue that wording limitations preclude a greater description of the context of six case sites. My suggestion is to cull the detail in the quantitative tables of admission rates, to show why the six sites were selected. While these are interesting they do not add data to the core argument of the paper. This would then free up space for more relevant detailed contextual description of the cases.

DATA COLLECTION

The data were collected from 142 informants through two rounds of interviews and focus groups. First from high level informants and then second from service delivery level staff. For a qualitative study this is a very large number of informants from which a huge volume of verbatim transcripts would be generated. Data that are presented in the paper are clearly a small selection of quotes from a small selection of informants. There appears to be no differentiation between those informants interviewed and those in focus groups nor differentiation of informants by their background, experience and knowledge of their local health system. Clearly the value of what informants say (that is their information richness) will not be the same, both because of who they are and what data collection method was used.

While there was a very good range of informants in the study, there is no information about how they were recruited, how many were approached and how many declined to participate.

I also note that there were 92 informants from the improving sites, but 51 informants from the deteriorating sites. Does this mean that the researchers had more data from the improving sites than the deteriorating sites?

It is stated that topic guides were used for interviews and focus groups, but no details about these guides is provided.

ANALYSIS

Three stages of analysis are described, first open coding, then the writing of case reports and finally analysis against the McKinsey 7S framework. It is not clear to me how these three stages were brought together to produce the findings. I would have liked to see some more description of the McKinsey's 7S framework to illustrate why they chose this, against which to "frame" the analysis.

The researchers state the following, about coding: "In line with the principle of constant comparison, each category was systematically checked for its internal consistency and interrelationships." I would like to know what the researchers did to conduct this systematic check.

I do not think that it works analytically to present four of McKinsey's categories together in one section (shared values,

	style, staff and skills). These are different concepts and hence I did not find this section to be analytically coherent. Given that the researchers did not fully use separately each of the 7S of McKinsey framework, and they suggest that other frameworks might be better suited to whole system analysis, such as Beers Viable Systems Model, then I suggest they redo the analysis using a more suited framework. I would think that the researchers have plenty of open coded data and case reports to use to rerun such an analysis. Given the large volume of transcript data that would be expected from 142 informants, but the small number of informant quotes provided, how can the reader be assured that the researchers' have not "cherry picked" selected quotes to show that improving sites had better systems in place. The more detailed case study approach I mention above (six cases) would at least enable the researchers to look for variation in each case both across and within the improving and deteriorating categories. That is to look for the deviant case (or outlier). This would give the reader the confidence that the researchers are not just cherry picking in a self-fulfilling bias.
--	---

REVIEWER	Mome Mukherjee Usher Institute of Population Health Sciences and Informatics, The University of Edinburgh
REVIEW RETURNED	10-Oct-2018

GENERAL COMMENTS	This is a very good piece of work and it was a pleasure to read. But the authors do not clearly mention that a previous work was already carried by them, published (Health Services and Delivery Research. 2015;3:37) and had similar findings, though they cite it: https://www.ncbi.nlm.nih.gov/books/NBK311369/ . What is new is I suppose two years of additional data, which did not change the findings. Could the authors justify what this manuscript adds to the knowledge of health services research? Minor points :  i) It would have helped if the 7S themes were articulated in the manuscript. ii) The COREQ checklist was not mentioned in the manuscript. iii) Pg 4/22, line 26 check "... although In England ..." iv) Pg 6/22 line 23 check "... University xx ..."
---

REVIEWER	Seamus Kent University of Oxford, United Kingdom
REVIEW RETURNED	02-Dec-2018

GENERAL COMMENTS	I found this interesting study to be well conducted and clearly written. There are important limitations to the analysis including the selection of groups based on recent performance combined with the retrospective justification for the performance, the suggestion of some regression to the mean as indicated by the greater similarity in trends after 3 years, and the fact that the analysis period fell within a time of significant institutional change. However, all these limitations are appropriately acknowledged in the Discussion.
---

	Often the authors refer to the “improving” sites. However, over five years, the average increase in the admission rate is actually positive. I understand that they may be performing better than average, but the use of the word “improving” may be misleading. I recommended revising this. The selection criteria for the 3 “improving” and 3 “deteriorating” sites were clear but how the specific cases were selected was less clear. Where they similar the highest ranked who met the criteria? The authors briefly state that data from 2007/8 was adjusted between selection and analysis periods. Comparing tables 1 and 2, these changes were actually very large, and it might be worth clarifying what changes were made and why. Furthermore, does using the updated figures change the ranking of studies by trends in admission rates? It seems from Table 2 that these studies are no longer “improving”. This should perhaps be included as an additional limitation.
--	---

REVIEWER	Koen Van den Heede KULeuven - Belgium
REVIEW RETURNED	02-Dec-2018

GENERAL COMMENTS	This topical paper is well written and reports about a well executed mixed methods design. It gives insight in which system components are important in the reduction of unplanned admissions in the population of 85+. My comments are of minor nature and I suggest to accept this manuscript when the authors integrate these comments: 1) introduction and discussion. Both sections are entirely focused on England. As this topic is not limited to the English context I suggest to rewrite both sections to make them more interesting for the international audience. By integrating references from other countries and rewriting these sections this could be resolved 2) Methods Add the main topics of the topic guide or add some examples 3) Table 1 D3 is still performing better than I1 and I3 at the latest year. Is it possible that the intervention sites are more successful because of a worse baseline? This finding is nowhere discussed in the paper 4) declining participants Not entirely clear what the reason of declination was. The authors might also elaborate on how they approached the sites. 5) Results Please add some examples about what is understood as being 'intermediate care'
---

REVIEWER	Dr Ian Pope Norfolk and Norwich University Hospital, UK
REVIEW RETURNED	05-Dec-2018

GENERAL COMMENTS	Thank you for asking me to review this excellent manuscript.
--

	This is a study of organisational factors impacting upon admission rates for those aged 85+. It is well designed, conducted and presented. Below are my comments and suggestions. Page 2, line 29: I struggle to understand what you mean by the although in "Although there were differences within the two groups of sites...". The results section of the abstract in general is slightly difficult to read and could be presented in a more logical way. Page 3, lines 7-15: Each of these feels like sentences so should have a full stop. Page 5, line 27: Please explain why you only covered the 5 years for study sites rather than for the selection of site. Page 5, line 45: in-depth rather than in depth. Page 5, line 47: either capitalise Emergency Departments (my preference) or uncapitilise acute medical units. Page 6, line 23: "University xx" I presume this has been done for anonymity but just wanted to make sure you intend to correct before publication Page 6, line 40: Please explain which classification and list of ambulatory care sensitive conditions you are using so readers can know whether you mean emergency conditions which don't require admission or chronic conditions that if well managed avoid the need for admissions. Page 10, line 23 & 24: "...," to "..." and ".." to "..." Page 10, line 30: close quotation marks required Page 18, line 22: move declined up a line My general comments are below: More explanation of inclusion or exclusion of blinding is required. Whilst I understand that it was useful for the interviewers to be unblinded, I would have thought it would have been helpful to have someone blinded to the site category do a round of analysis in order to ensure bias wasn't being introduced. I was very surprised that there was so little discussion of the role of the ambulance services given they represent a key opportunity to avoid conveyance to hospital. It would be helpful to provide either details if they were involved or explanation if not.
--	---

VERSION 1 – AUTHOR RESPONSE

REVIEWER 1 The stated method is case study, but I am unclear what are defined as the cases. Are these each of the six sites (that is six case studies) or are these the two groups of three sites (that is two case studies). I raise this as the results are presented thematically with the differentiation made cross-sectionally according the category "improving sites" or "deteriorating sites". This would suggest two case studies. However, in describing the analysis, the authors refer to the middle analytic step of writing up six case reports, which suggest six case studies. In my view a method of using six case	The reviewer is correct that the study comprised six in-depth case studies, where the unit of analysis was the care system within which urgent care was managed. We concur that this affords more rigorous and comparative analysis of the factors shaping the management and provision of unplanned urgent care for people over 85 years.
---	---

studies would be a more rigorous study design as this would lend itself to the use of greater contextual description of each case, including the differences in populations being served. Six site case studies would also lend itself to the search for variation and disconfirming findings across sites within the improvement and deteriorating categories	We acknowledge the reviewer's observation that this is not necessarily clear from the way the data is presented, and that some of the wider contextual detail is missing (see following comment). To this end we have amended Table 3 to include more contextual information about each case study system.
At present there is very little contextual description provided other than what is provided in the selective quotes and in the overall statements made by the authors about the differences between the improving and deteriorating site categories collectively. The reader is left to accept these author descriptions at face value as accurate differentiations. The strength of case study as a research design is the detailed use of context to help explain variation, and so more detailed case context would strengthen the veracity of the claims being made in the paper. The authors may argue that wording limitations preclude a greater description of the context of six case sites. My suggestion is to cull the detail in the quantitative tables of admission rates, to show why the six sites were selected. While these are interesting they do not add data to the core argument of the paper. This would then free up space for more relevant detailed contextual description of the cases.	The amendments to Table 3 now provide additional contextual information about each case study site. We have also created a new table (Table 4) that further summarises each case study according to the McKinsey 7S model, and which enables more direct comparison between sites. We considered culling the quantitative data but note that other reviewers have found them helpful in interpreting site selection and interpretation of results.
The data were collected from 142 informants through two rounds of interviews and focus groups. First from high level informants and then second from service delivery level staff. For a qualitative study this is a very large number of informants from which a huge volume of verbatim transcripts would be generated. Data that are presented in the paper are clearly a small selection of quotes from a small selection of informants. There appears to be no differentiation between those informants interviewed and those in focus groups nor differentiation of informants by their background, experience and knowledge of their local health system. Clearly the value of what informants say (that is their information richness) will not be the same, both because of who they are and what data collection method was used.	We have added the role of the informant for each quote and noted that all quotes selected came from one to one interviews.
While there was a very good range of informants in the study, there is no information about how they were recruited, how many were approached and how many declined to participate.	Yes, more participants were recruited from improving than deteriorating sites, and this is now noted in the findings section. We acknowledge that this might

I also note that there were 92 informants from the improving sites, but 51 informants from the deteriorating sites. Does this mean that the researches had more data from the improving sites than the improving sites?	give greater depth of insight from the improving sites, but as detailed in Table 3 recruitment varied between sites, with variations in recruit for both improving (15-43) and declining (between 7-24) sites. Given the way data was analysed both within and between case we do not think this variation has significantly biased the study.
It is stated that topic guides were used for interviews and focus groups, but no details about these guides is provided.	The main headings of the topic guide are now described in the methods section and a full topic guide presented as an appendix.
Three stages of analysis are described, first open coding, then the writing of case reports and finally analysis against the McKinsey 7S framework. It is not clear to me how these three stages were brought together to produce the findings.	More detail has been added to the analysis section.
I would have liked to see some more description of the McKinesy's 7S framework to illustrate why they chose this, against which to "frame" the analysis.	We have now included figure 1, which explains the framework and how we interpreted it.
The researchers state the following, about coding: "In line with the principle of constant comparison, each category was systematically checked for its internal consistency and interrelationships." I would like to know what the researchers did to conduct this systematic check	Each (open coded) category of data was assigned a brief description, detailing what was assumed or interpreted by the researchers to be significant about the section of data, and then when carrying out framework analysis how it related to the McKinsey 7S framework. These codes and descriptions provided the basis of regular discussion amongst research team, first for reviewing the consistency of the analysis process (inter-coder), i.e. so that similar codes related to similar phenomena. As part of this process the similarities and differences between codes were analysed, especially when relating data the 7S model, to ensure that data/codes were categorised in ways that were sufficiently distinct, or where they shared common or complementary features they were aggregated into higher order codes, which was especially important when relating the data to the McKinsey model.
I do not think that it works analytically to present four of McKinesy's categories together in one section (shared	All seven categories are now presented separately in table 4, but given the

values, style, staff and skills). These are different concepts and hence I did not find this section to be analytically coherent. Given that the researchers did not fully use separately each of the 7S of McKinsey framework, and they suggest that other frameworks might be better suited to whole system analysis, such as Beers Viable Systems Model, then I suggest they redo the analysis using a more suited framework.	relative paucity of information for and overlap between shared values, style, staff and skills, these are combined in the text. We acknowledge in the discussion that the 7S framework worked better for some components than others. It is not feasible to reanalyse our data at this stage using a different framework because the 7S framework was partially foundational in defining the methodology at the start of the work. The framework was used to help identify/segregate system parts, then capture data, then followed by analysis. The Stafford Beer Viable System Model (VSM) is indeed good for looking at complex systems but is greatly limited by its 'accessibility' – it is difficult to understand and use by anyone who is not trained in systems engineering thinking. Additionally, the VSM it is not as intuitive as the 7S framework for communicating its outcomes, via the 7 framework components, to all stakeholders.
Given the large volume of transcript data that would be expected from 142 informants, but the small number of informant quotes provided, how can the reader be assured that the researchers' have not "cherry picked" selected quotes to show that improving sites had better systems in place.	We have now stated that additional quotes are presented in the funders' report, referenced in the paper and added more detail about analysis.
REVIEWER 2	
This is a very good piece of work and it was a pleasure to read. But the authors do not clearly mention that a previous work was already carried by them, published (Health Services and Delivery Research. 2015;3:37) and had similar findings, though they cite it: https://www.ncbi.nlm.nih.gov/books/NBK311369/ . What is new is I suppose two years of additional data, which did not change the findings. Could the authors justify what this manuscript adds to the knowledge of health services research	Thank you This paper is a summary of qualitative findings of the mixed method study, the full report of which has been published by NIHR. No new data are presented here. This has been made clearer in the last sentence of the introduction.
It would have helped if the 7S themes were articulated in the manuscript.	These are now described in figure 1.
The COREQ checklist was not mentioned in the manuscript.	The COREQ checklist was submitted with the manuscript

Pg 4/22, line 26 check "... although In England ..."	Corrected
Pg 6/22 line 23 check "... University xx ..."	Corrected
REVIEWER 3	
I found this interesting study to be well conducted and clearly written	Thank you
There are important limitations to the analysis including the selection of groups based on recent performance combined with the retrospective justification for the performance, the suggestion of some regression to the mean as indicated by the greater similarity in trends after 3 years, and the fact that the analysis period fell within a time of significant institutional change. However, all these limitations are appropriately acknowledged in the Discussion.	Thank you
Often the authors refer to the "improving" sites. However, over five years, the average increase in the admission rate is actually positive. I understand that they may be performing better than average, but the use of the word "improving" may be misleading. I recommended revising this.	We feel this descriptor, although not ideal, is appropriate as rates were 'improving' or 'deteriorating' at the time they were selected. We have added a sentence in the strengths and limitations section (3rd para) noting that over a five year period two of the improving sites showed a small overall increase in admission rates, although this was less than in deteriorating sites.
The selection criteria for the 3 "improving" and 3 "deteriorating" sites were clear but how the specific cases where selected was less clear. Where they similar the highest ranked who met the criteria?	Yes, this has been clarified (2 nd para site selection)
The authors briefly state that data from 2007/8 was adjusted between selection and analysis periods. Comparing tables 1 and 2, these changes were actually very large, and it might be worth clarifying what changes were made and why. Furthermore, does using the updated figures change the ranking of studies by trends in admission rates? It seems from Table 2 that these studies are no longer "improving". This should perhaps be included as an additional limitation.	As above, we have added a sentence to this effect in the strengths and limitations section
REVIEWER 4	
This topical paper is well written and reports about a well executed mixed methods design. It gives insight in which system components are important in the reduction of unplanned admissions in the population of 85+. My comments are of minor nature and I suggest to accept this manuscript when the authors integrate these comments:	Thank you

1) introduction and discussion. Both sections are entirely focused on England. As this topic is not limited to the English context I suggest to rewrite both sections to make them more interesting for the international audience. By integrating references from other countries and rewriting these sections this could be resolved	We have added a sentence at the beginning of the introduction emphasising that this is an international problem, but given the setting of the study and word count constraints, we feel it is reasonable to focus on England in the discussion
2) Methods Add the main topics of the topic guide or add some examples	Added (see earlier response)
3) Table 1 D3 is still performing better than I1 and I3 at the latest year. Is it possible that the intervention sites are more successful because of a worse baseline? This finding is nowhere discussed in the paper	In the penultimate para of the introduction we explain why our focus is on changes rather than absolute rates. We have added the possibility of regression to the mean in the discussion.
4) declining participants Not entirely clear what the reason of declination was. The authors might also elaborate on how they approached the sites.	We have added more detail on how participants were approached at the end of the 'site selection' section.
5) Results Please add some examples about what is understood as being 'intermediate care'	We have clarified that intermediate care may have been home or institution based.
REVIEWER 5	
Thank you for asking me to review this excellent manuscript. This is a study of organisational factors impacting upon admission rates for those aged 85+. It is well designed, conducted and presented.	Thank you
Page 2, line 29: I struggle to understand what you mean by the although in "Although there were differences within the two groups of sites...". The results section of the abstract in general is slightly difficult to read and could be presented in a more logical way.	We agree this phrase is unclear and have reworded the results section of the abstract.
Page 3, lines 7-15: Each of these feels like sentences so should have a full stop.	Added
Page 5, line 27: Please explain why you only covered the 5 years for study sites rather than for the selection of site.	We feel this is explained by the following: 'After the sites had been selected we used updated HES data to examine 85+ admission rates over five years (2007/08 to 2011/12) rather than just the three years used for selection' (last para of site selection).
Page 5, line 45: in-depth rather than in depth.	Corrected
Page 5, line 47: either capitalise Emergency Departments (my preference) or uncapitilise acute medical units.	Corrected

Page 6, line 40: Please explain which classification and list of ambulatory care sensitive conditions you are using so readers can know whether you mean emergency conditions which don't require admission or chronic conditions that if well managed avoid the need for admissions.	This has been clarified and a reference added (both types were considered)
Page 10, line 23 & 24: "...," to "... " and ". ." to "... "	done
Page 10, line 30: close quotation marks required Page 18, line 22: move declined up a line	done
More explanation of inclusion or exclusion of blinding is required. Whilst I understand that it was useful for the interviewers to be unblinded, I would have thought it would have been helpful to have someone blinded to the site category do a round of analysis in order to ensure bias wasn't being introduced.	The following sentence has been added to the strengths and limitations section: 'In retrospect, blinding some rounds of data analysis to categorisation of sites may have reduced the risk of bias'.
I was very surprised that there was so little discussion of the role of the ambulance services given they represent a key opportunity to avoid conveyance to hospital. It would be helpful to provide either details if they were involved or explanation if not.	The following sentence has been added to the strengths and limitations section: 'Finally, the snowball sampling technique may have led to under-representation of some groups, such as ambulance services'.

VERSION 2 – REVIEW

REVIEWER	Jeffrey Fuller Flinders University, Australia
REVIEW RETURNED	26-Feb-2019

GENERAL COMMENTS	The authors have sufficiently addressed my critique. The acronym PPI involvement appears on p6 - but I don't think it has been fully spelt out prior to this. I accept that the McKinsey 7S remain as the analytic framework despite its limitations. I am not convinced that values, skills, style and staff are overlapping and while table 4 helps to understand something of these categories, without the framework definition of each of the 7 S's it is a bit difficult to know. I am not suggesting that you add these definitions (although you could as an appendix), but given that you place some material related to staffing and skills elsewhere, consider relabelling section 4. In fact most of section 4 is described by the authors as under shared values.
---

REVIEWER	Mome Mukherjee The University of Edinburgh, UK
REVIEW RETURNED	12-Mar-2019

GENERAL COMMENTS	Although this paper now says that it focuses on the qualitative findings of the previous work (https://www.ncbi.nlm.nih.gov/books/NBK311369/), but it is reporting on the same data and same findings. I cannot find how this fits the bill for a new publication. There are several concerning issues in this manuscript:  i) While they mention the objective was to find out about “unplanned” admissions in 85+, nowhere in the title is that made clear. ii) There were 142 participants in interviews and focus groups and the topic guide is very long, which must have contributed to enormous amount of data. But data were not presented by care sectors or type of health /professionals, which would have been interesting. iii) It seems the Framework Approach was used to fit data to the pre-defined 7S categories, rather than doing any independent thematic analyses. No further information is provided on themes, sub-themes, using quotes. It seems the authors cherry picked the quotes to make the points around the pre-decided 7S categories. iv) Out of 7S, four Ss are reported together, which is surprising. v) Although PPI was involved, I could not find out what their views were on the study topic. vi) I could not find number of 85+ people or cases in each of those sites. Only percentages or rates are given, unless I have missed. Since the percentage differences are narrow between “improving” and “deteriorating” sites, based on which the study has been done, corresponding numbers are very important to have to check if very small numbers are causing the change in rates or if rates are based on large number of cases. vii) Study period was not mentioned. viii) PCTs were ranked based on data of 2007-09, which is 10 or more years old. Why more recent data were not used is not mentioned anywhere. One would expect changes in these 10 years and more. ix) Although “deprivation” is mentioned in Table 3, nowhere else is it found in the manuscript. We know deprivation is an important factor associated with illness. The authors do not mention how deprivation could have been associated with the performance of the sites. x) The authors compare the two groups- “improving” and “deteriorating” sites. Besides the similar statistical ranking, one would expect there are differences in each of these sites within the groups they belong. Such has not been discussed. xi) While the target of the paper is towards reducing unplanned hospital admissions, but we find the elderly are too frail to look for care outside the usual places of care and care organisations are disjointed. Given these circumstances, it is worrying that the authors do not reflect on how the care system could be integrated, so that by reduction of hospital admissions of the elderly we do not end up in not caring for them. This is further concerning with the PPI voice missing in the reporting. Minor points
---

	xii) It would have helped if the manuscript was reported as per COREQ checklist, also to defy that it was not submitted perfunctorily. xiii) There are two Appendix 2. xiv) Please try to fit individual tables in one page for Tables 3, 4 xv) Pg 45, line 4, "... for 2010/11 and 2011 showed ..." – please check xvi) Citations are sometimes within sentences and sometimes after full stop –please adhere to journal guidelines. xvii) Definition of some S of 7S are in the Results. They should be in Methods. xviii) Although the manuscript says its focussed on the qualitative findings, there are reporting on quantitative findings as well. xix) Page 38, line 47 –check "... subsequence results section,..."
--	--

REVIEWER	Seamus Kent University of Oxford, UK
REVIEW RETURNED	15-Feb-2019

GENERAL COMMENTS	I am happy with the amendments made to the paper following my previous comments.
--

REVIEWER	Koen Van den Heede KCE (Belgium)
REVIEW RETURNED	16-Feb-2019

GENERAL COMMENTS	the comments on a previous version were integtraed in the revised manuscript in a satisfactory manner
---

REVIEWER	Dr Ian Pope Norwich Medical School University of East Anglia Norwich UK
REVIEW RETURNED	20-Feb-2019

GENERAL COMMENTS	I am satisfied with the responses to my comments and am happy with the revised manuscript.
--

VERSION 2 – AUTHOR RESPONSE

R1	
The acronym PPI involvement appears on p6 - but I don't think it has been fully spelt out prior to this.	This has now been spelt out (P6, L5)
I accept that the McKinsey 7S remain as the analytic framework despite its limitations. I am not convinced that values, skills, style and staff are overlapping and while table 4 helps to	We agree with this suggestion and have retitled section 4 as Shared Values, with an explanation of the remaining categories at the end of the results section (P11, L37-40)

understand something of these categories, without the framework definition of each of the 7 S's it is a bit difficult to know. I am not suggesting that you add these definitions (although you could as an appendix), but given that you place some material related to staffing and skills elsewhere, consider relabelling section 4. In fact most of section 4 is described by the authors as under shared values.	
Although this paper now says that it focuses on the qualitative findings of the previous work (https://eur03.safelinks.protection.outlook.com/?url=https%3A%2F%2Fwww.ncbi.nlm.nih.gov%2Fbooks%2FNBK311369%2F&data=02%7C01%7Caw7%40leicester.ac.uk%7C7b7d285273a54215cb3608d6d525bed7%7Caebcd6a31d44b0195ce8274afe853d9%7C0%7C0%7C636930755431575884&sdata=SKCVXetVtz8JaNUg08x%2F3b8xx4p1gCUuzFqOSpolqY%3D&reserved=0), but it is reporting on the same data and same findings. I cannot find how this fits the bill for a new publication.	This link is to the project report published by the NIHR Health Service Delivery Research. This does not constitute prior publication.
R2	
i) While they mention the objective was to find out about “unplanned” admissions in 85+, nowhere in the title is that made clear.	Thank you. The title has been amended, replacing ‘acute’ with ‘unplanned’.
ii) There were 142 participants in interviews and focus groups and the topic guide is very long, which must have contributed to enormous amount of data. But data were not presented by care sectors or type of health /professionals, which would have been interesting.	In response to the first round of reviews, we added the role of the informant following each quote. This includes information about care sector and organisation, and where relevant, profession.
iii) It seems the Framework Approach was used to fit data to the pre-defined 7S categories, rather than doing any independent thematic analyses. No further information is provided on themes, sub-themes, using quotes. It seems the authors cherry picked the quotes to make the	We chose to fit the data using a framework approach (in this case the 7S) for analysis, rather than an inductive approach. However as stated in the methods section (P6, L27-30), within each category we applied the constant comparison method, including searching for counterfactuals.

points around the pre-decided 7S categories.	
iv) Out of 7S, four Ss are reported together, which is surprising.	This has been amended, as above.
v) Although PPI was involved, I could not find out what their views were on the study topic.	Several PPI focus groups were held, as shown in table 3, and findings are presented in the project report. However in the context of this paper, quotes from these groups did not contribute to the themes presented.
vi) I could not find number of 85+ people or cases in each of those sites. Only percentages or rates are given, unless I have missed. Since the percentage differences are narrow between “improving” and “deteriorating” sites, based on which the study has been done, corresponding numbers are very important to have to check if very small numbers are causing the change in rates or if rates are based on large number of cases.	A column has been added to Table 3 showing number of people aged 85+ and this as a % of the total population
vii) Study period was not mentioned.	This has been added (P6, L11)
viii) PCTs were ranked based on data of 2007-09, which is 10 or more years old. Why more recent data were not used is not mentioned anywhere. One would expect changes in these 10 years and more.	When the study started in 2012, and sites were being identified, data from 2009 were the latest available. More recent data were sought from each site after participation had been confirmed.
ix) Although “deprivation” is mentioned in Table 3, nowhere else is it found in the manuscript. We know deprivation is an important factor associated with illness. The authors do not mention how deprivation could have been associated with the performance of the sites.	This has been added as an example of a ‘relatively immutable factor’ to justify out method of examining changes in admission within sites.
x) The authors compare the two groups- “improving” and “deteriorating” sites. Besides the similar statistical ranking, one would expect there are differences in each of these sites within the groups they belong. Such has not been discussed.	Differences between sites are presented in the tables and discussed in the ‘strengths and limitations’ section (P13, L 8-9)

xi) While the target of the paper is towards reducing unplanned hospital admissions, but we find the elderly are too frail to look for care outside the usual places of care and care organisations are disjointed. Given these circumstances, it is worrying that the authors do not reflect on how the care system could be integrated, so that by reduction of hospital admissions of the elderly we do not end up in not caring for them. This is further concerning with the PPI voice missing in the reporting.	We believe we have emphasised the importance of integrated services in the discussion, for example the importance of integrate intermediate care (P12, L9)
xii) It would have helped if the manuscript was reported as per COREQ checklist, also to defy that it was not submitted perfunctorily.	I don't fully understand this comment; The COREQ checklist was included at submission.
xiii) There are two Appendix 2.	This has been corrected: there is only one appendix
xiv) Please try to fit individual tables in one page for Tables 3, 4	I presume this can be done during typesetting, the tables are too large to do this in Word
xv) Pg 45, line 4, "... for 2010/11 and 2011 showed ..." – please check	Corrected to 2011/12
xvi) Citations are sometimes within sentences and sometimes after full stop –please adhere to journal guidelines.	We feel in some cases it is preferable to present the citation mid-sentence so it clearer to what issue it refers, but would be happy to amend if this is not journal policy.
xvii) Definition of some S of 7S are in the Results. They should be in Methods.	In the methods section, reference is made to figure 1, which defines 7S.
xviii) Although the manuscript says its focussed on the qualitative findings, there are reporting on quantitative findings as well.	We believe the quantitative findings enable the qualitative findings to be seen in context.
xix) Page 38, line 47 –check "... subsequence results section,..."	corrected